# Perspectives on preconception care in Ethiopia: Social, cultural, and structural determinants

**Yared Asmare Aynalem**[1,2]*, **Pauline Paul**[1], **Zohra S. Lassi**[3,4], **Salima Meherali**[1]

**1** Faculty of Nursing, University of Alberta, Edmonton, Alberta, Canada, **2** College of Health Science, Debre Berhan University, Debre Berhan, Ethiopia, **3** School of Public Health, The University of Adelaide Faculty of Health and Medical Sciences, Adelaide, South Australia, Australia **4** Robinson Research Institute, Adelaide University, Adelaide, South Australia, Australia

☙ These authors contributed equally to this work.

* aynalem@ualberta.ca

## Abstract

### Background

Although Ethiopia introduced its first national preconception care (PCC) guideline in 2024, PCC remains rarely integrated into routine practice, and existing studies have largely focused on women's knowledge and behaviors. Little is known about how adults navigate PCC within broader social, cultural, and structural contexts. This study provides an in-depth urban Ethiopian analysis of how adults experience and negotiate PCC within intersecting gender, moral, and institutional systems, offering insights beyond individual-level understanding.

### Methods

An interpretive description design guided semi-structured interviews with 18 adults (10 women, 8 men; 19–45 years) recruited through maximum-variation sampling from two public hospitals in Addis Ababa. Interviews were conducted in Amharic, transcribed, translated, and analyzed inductively. Data analysis was guided by ID principles, complemented by thematic analysis techniques informed by grounded theory, including line-by-line coding, constant comparison, and analytic memoing. Field notes captured contextual and relational dynamics.

### Results

Seven interrelated themes highlighted complex dynamics in PCC. Knowledge was fragmented and often recognized only after complications, shaped by marital gatekeeping, faith-based beliefs, and exclusion of unmarried women. PCC was valued as protective and morally significant, but stigma, poverty, staff shortages, and inconsistent services constrained practice. Men were largely financial supporters, though many expressed a desire to participate, limited by gender norms and

**Data availability statement:** All relevant data are within the paper and its Supporting Information file.

**Funding:** YAA received the Vanier Canada Graduate Scholarship (Government of Canada; https://vanier.gc.ca) and the International Development Research Centre (IDRC) Doctoral Research Award (https://idrc-crdi.ca). The funders had no role in study design, data collection and analysis, decision to publish, or preparation of the manuscript.

**Competing interests:** NO authors have competing interests.

women-centered services. Pharmacies and digital media provide informal but sometimes unsafe guidance. Emotional experiences, fear, guilt, secrecy, and hope were central to PCC engagement. Education, peer influence, schools, and community leaders emerged as catalysts for uptake, yet participants emphasized that sustainable PCC required visible institutional support, reliable services, and government recognition. Strategies to enhance practice included simplifying communication, creating accessible clinic entry points, and mobilizing community networks to normalize pre-pregnancy preparation.

## Conclusions

This study reveals PCC in urban Ethiopia as a socially negotiated, morally contested, and structurally uneven practice, far more complex than knowledge deficits imply. These findings offer novel, actionable direction for implementing Ethiopia's PCC guideline through visible, inclusive, relational, and community-anchored approaches that address the social conditions shaping PCC access.

## Introduction

Maternal and child health remains central to the Sustainable Development Goals (SDGs), which aim to reduce the global maternal mortality ratio to fewer than 70 per 100,000 live births and to end preventable newborn deaths by 2030 [1,2]. However, progress remains insufficient. In 2023, an estimated 260,000 women died from pregnancy-related causes, corresponding to a global maternal mortality ratio of approximately 197 per 100,000 live births, nearly three times the SDG target [2]. Each year, about 2.3 million newborns die within the first month of life, with most of these deaths occurring in low- and middle-income countries [3]. These preventable losses highlight that health trajectories are shaped well before conception and that interventions limited to antenatal care are often too late to avert complications [4,5]. To address this gap, the World Health Organization (WHO) defines preconception care (PCC) as a package of biomedical, behavioral, and social interventions delivered before conception to optimize parental health and improve pregnancy outcomes [6]. In 2024, WHO renewed its call to integrate PCC within national health systems as a proactive strategy to prevent complications and break intergenerational cycles of inequity [7]. This reflects recognition that pregnancy outcomes are influenced not by isolated events but by lifelong exposures, behaviors, and social determinants [8].

The benefits of PCC are well established. Preconception folic acid and multivitamin supplementation reduce neural tube defects and other adverse pregnancy outcomes [9]. Broader interventions, such as improved nutrition, screening and management of chronic conditions, and reduction of harmful exposures, enhance fertility, maternal health, and neonatal survival [5,10]. Because these actions precede pregnancy, PCC is consistently more cost-effective than antenatal interventions [11] and has been described as the "missing link" in maternal and child health strategies [12]. While PCC is intended to involve both women and men, men's participation has

often been neglected. The Developmental Origins of Health and Disease (DOHaD) framework shows that maternal and paternal health before conception influences offspring outcomes across the life course [13]. Men's health factors, including obesity, diabetes, smoking, alcohol use, and micronutrient deficiencies, affect fertility, miscarriage, and congenital anomalies [14]. Beyond biology, men often shape reproductive decisions and household resources, making their exclusion a barrier to equitable and effective PCC [15].

In Ethiopia, the need for PCC is urgent. The maternal mortality ratio remains 267 deaths per 100,000 live births, with preventable conditions such as anemia, hypertensive disorders, and congenital anomalies continuing to undermine survival [2,16]. Although family planning and antenatal services have expanded, PCC has historically been absent from national policies, health information systems, and routine delivery [17]. The launch of Ethiopia's first National PCC Guideline in 2024 marked a significant milestone [18], but implementation remains weak. Awareness and utilization are low, often limited to folic acid supplementation, and men's involvement remains minimal and stigmatized [19,20]. Weak institutional integration, inadequate training, and poor supervision further constrain delivery [21,22]. These barriers intersect with gendered and socio-cultural norms. In many communities, PCC is framed narrowly, and stigma, poverty, and gender expectations determine who is considered eligible for care [4,23]. Most Ethiopian studies are quantitative and focus on women's awareness or utilization, often attributing low uptake to knowledge deficits [24]. Such approaches risk oversimplifying complex realities and highlight the need for qualitative, context-sensitive inquiry [4,25].

Given this gap, the present study uses an Interpretive Description (ID)design [26] to explore the perspectives and experiences of PCC among reproductive-age adults (18–49 years) in Addis Ababa. By situating participants' perspectives within socio-ecological and intersectional contexts, the study aims to generate insights to inform culturally responsive services and guide policy for integrating PCC into Ethiopia's reproductive health continuum.

## Methods

### Design and methodological orientations

This study forms part of a larger qualitative interpretive project that explored perspectives on PCC among multiple groups, including healthcare workers, stakeholders, and reproductive-age individuals. The present paper focuses specifically on the perspectives and experiences of reproductive-age individuals. The study employed an interpretive descriptive (ID) qualitative design, a flexible yet rigorous approach suited to applied health research [26]. Unlike approaches oriented towards theoretical abstraction or the distillation of experience in isolation from its context, ID is explicitly designed to generate clinically relevant insights that can inform practice and policy. It allows researchers to engage with existing contextual knowledge as part of the interpretive process rather than setting it aside, and to identify both patterns of commonality and meaningful variation across participants' accounts. This made ID particularly appropriate for examining how adults of reproductive age in Ethiopia understand, experience, and navigate PCC, where institutional, cultural, and structural forces are deeply embedded in how individuals make sense of and engage with reproductive health services, and where findings need to be actionable within that specific context

### Research questions

1. What are the perspectives and experiences of reproductive-age men and women (18–49 years) regarding preconception care (PCC)?

2. How do socio-ecological factors and intersecting determinants, such as gender, socioeconomic status, cultural beliefs, and healthcare accessibility, shape these perspectives and experiences of PCC among reproductive-age individuals (18–49 years)?

## Study setting and participant recruitment

The study was conducted in two major public hospitals in Addis Ababa. These hospitals were purposively selected because they are among the largest public facilities providing comprehensive reproductive and maternal health services and serve patients from diverse socioeconomic and geographic backgrounds across the city. Their high patient volumes and wide range of services (including antenatal, postnatal, family planning, and general outpatient care) made them appropriate settings to capture varied perspectives from individuals at different stages of the reproductive life course. Eligible participants were adults aged 18–49 years who lived in Addis Ababa and accessed services at these hospitals. To ensure diverse perspectives, recruitment targeted men and women with varying marital, educational, employment, and reproductive backgrounds, including pregnant individuals, partners of pregnant women, those planning to conceive, individuals with recent pregnancy experience, and those with fertility concerns. Participants were recruited across antenatal, family planning, postnatal, and outpatient units using purposive maximum variation sampling, supported by hospital staff, posters and flyers, and verbal invitations in Amharic. Women planning to conceive were identified through collaboration with healthcare workers during routine family planning or reproductive health visits, based on their perspective of patients' reproductive intentions. Written informed consent was obtained from all participants. Maximum variation sampling was appropriate for this interpretive descriptive study because it enabled the capture of diverse experiences and contextual differences, supporting in-depth exploration of PCC perspectives.

## Data collection

Data collection was conducted between 31/03/2025 and 28/08/2025 through in-depth, semi-structured interviews with 18 reproductive-age individuals (10 women and 8 men). Guided by the interpretive descriptive design [26], interview guides were developed through literature review, expert consultation, and alignment with study objectives, with intersectionality theory and the socio-ecological model informing their design to capture overlapping identities and multilevel influences (S1 File). The guides explored the perspectives, perceptions, and experiences of PCC; perceived benefits and barriers; gender and cultural dynamics; and experiences with healthcare providers, while maintaining flexibility to allow participants to raise issues they considered important. To enhance rigor and cultural relevance, the guides were pilot tested with three individuals (2 women and 1 man) similar to the study population but not included in the main analysis. The pilot assessed clarity, comprehensibility, and appropriateness of questions, as well as the flow and timing of the interview, which led to refinements in wording, sequencing, and probing strategies. These adjustments strengthened the guides' ability to elicit rich, nuanced data while foregrounding participants' perspectives, consistent with interpretive descriptive methodology. All interviews were conducted in Amharic by the principal investigator in private hospital spaces to ensure confidentiality and comfort, lasted 30–60 minutes, and were audio-recorded with written informed consent. Field notes captured contextual observations, participant demeanor, non-verbal cues, and early analytic reflections.

## Data analysis

Data analysis was guided by ID principles, which involve moving iteratively between data and emerging interpretations to construct a coherent and clinically relevant understanding of participants' experiences, complemented by thematic analysis techniques informed by grounded theory [26,27]. This hybrid approach supported inductive reasoning and analytic depth, generating insights closely tied to participants' lived experiences. All interviews were audio-recorded with participants' consent and transcribed verbatim in Amharic by the primary researcher. All transcripts were fully de-identified, with the removal of any potentially identifying information. Participants did not consent to public sharing of full transcripts; therefore, only de-identified excerpts are presented, and additional data may be made available to qualified researchers upon reasonable request, subject to ethical approval. Transcripts were then translated into English by the primary researcher, and translated transcripts were shared with participants for verification to ensure accuracy and preserve

cultural and linguistic meaning. Any discrepancies were resolved through discussion until a consensus was reached. Transcripts were imported into NVivo 15 for data management, coding, and comparative analysis. Data collection and analysis were conducted concurrently, allowing early insights to inform subsequent sampling and questioning, consistent with ID's emphasis on information power rather than saturation [26,28]. Sufficient information power was determined by considering the study aim, sample specificity, quality of dialogue, and analysis strategy. Recruitment continued until the data provided rich, detailed insights sufficient to address the research questions.

The analytic process began with immersion in the data through repeated readings and memo writing. Analysis was conducted by the primary researcher, with peer debriefing sessions held regularly with two co-authors to review emerging codes and interpretations, challenge assumptions, and enhance credibility. A codebook was developed inductively from initial open coding, capturing codes directly derived from participants' words, and was refined iteratively throughout the analytic process. Open coding was conducted inductively, generating concepts directly from participants' words. Codes were iteratively compared using the constant comparative method to identify similarities, contradictions, and variations both within and across interviews. Initial analysis was conducted separately by hospital and gender to examine intra-group patterns, followed by cross-group comparisons to identify convergences and divergences across gendered and institutional contexts. Axial coding explored relationships among codes and developed broader conceptual categories, such as gender roles, cultural norms, access to services, and perceptions of PCC. These categories were refined into central interpretive themes through selective coding, producing a cohesive narrative of participants' experiences. Reflexive memoing was maintained throughout to document interpretive decisions, monitor researcher positionality, and ensure transparency in the analytic process. Intersectional theory guided interpretation by examining how overlapping social identities, gender, age, marital status, and socioeconomic position shaped PCC knowledge, access, and engagement. The socio-ecological model informed the framing of findings across individual, interpersonal, community, and institutional levels, highlighting relational and systemic influences. Analytic synthesis was guided by core ID questions: What meanings are participants constructing about PCC? What is said, and what remains unspoken? How do gender, culture, and health system dynamics intersect to shape these perspectives? NVivo's matrix and case comparison tools supported systematic exploration, while writing memos documented interpretive decisions and reflexive positioning. The results were cohesively narrated, integrating open and axial codes into a coherent interpretive storyline that illuminates the complex, intersecting social, cultural, and systemic factors shaping PCC [26].

## Theoretical framework

Although this study did not adopt a single predefined theoretical framework to guide analysis, it was conceptually informed by the socio-ecological model and intersectionality theory [28,29]. The socio-ecological model highlights how health behaviors are shaped by interacting influences at individual, interpersonal, community, and institutional levels, while intersectionality theory explains how overlapping social identities, such as gender, age, marital status, and socioeconomic position, shape lived experiences and access to care [29,30]. Consistent with interpretive description, analysis was conducted inductively using thematic analysis informed by grounded theory techniques (open, axial, and selective coding), without imposing a priori theoretical structures. Instead, these frameworks informed the development of the semi-structured interview guide and supported the interpretation of findings. This approach aligns with Thorne's guidance that theoretical perspectives can be used to enhance interpretation without constraining the analytic process, thereby strengthening the connection between the findings and broader social and structural contexts [26].

## Rigor and trustworthiness

The primary researcher is a men health professional with a background in reproductive health and fluency in Amharic, the language in which all interviews were conducted. This positioning facilitated rapport and linguistic authenticity but also carried the potential for professional assumptions and interpretive bias. To manage this, the researcher maintained a reflexive

journal throughout data collection and analysis, documenting personal reactions, assumptions, and interpretive decisions. Prior relationships with participants were disclosed and managed through transparent communication of the research purpose, and care was taken to ensure participants felt free to express perspectives that diverged from clinical or institutional norms. Peer debriefing with co-authors provided an additional check on interpretive positioning.

Rigor was ensured using quality criteria specific to interpretive description [26]. Epistemological integrity was maintained by aligning the study's applied health purpose with its design, including interview guides linked to research objectives and analyses focused on practice-relevant insights. Representative credibility was achieved through purposive, maximum-variation sampling to capture diverse participant perspectives. Analytic logic was supported by systematic coding, memo writing, and regular supervisory feedback. Interpretive authority was strengthened through prolonged engagement, reflexive journaling, and triangulation of transcripts, field notes, and memos. Data collection was guided by information power rather than numerical saturation, concluding when sufficient depth and variation were achieved [26]. The study adhered to the COREQ checklist for reporting qualitative research (**S2 File**).

### Ethical considerations

Ethical approval was obtained from the University of Alberta Research Ethics Board (Pro00149685_AME1) and the Addis Ababa Health Bureau (Ref. No: አ/አ/ጤቢ/2/198/17), and permission to collect data was also secured from both hospitals. Written informed consent was obtained from all participants. Confidentiality was strictly maintained by de-identifying transcripts, securing data storage, and limiting access to the research team. Participants were informed of their right to withdraw at any time without consequences. All interviews were conducted with attention to cultural sensitivity, participant comfort, and ethical research practices.

## Result

### Participant characteristics

Eighteen participants (10 women, 8 men) aged 19–45 years were recruited from two selected public hospitals and represented varied marital, educational, and occupational backgrounds.

Their reproductive experiences, from pregnancy planning and recent births to fertility challenges, offered diverse perspectives on how PCC was understood, perceived, and practiced in daily life. Participants' characteristics are presented in **Table 1**

### Overview of findings

Seven themes illustrate how participants engaged with PCC: (1) Fragmented Knowledge Pathways and Practices across Reactive, Guided, and Constrained Routes; (2) Preventive Orientation and Reassurance; (3) Emotional Struggles and Coping Mechanisms; (4) Advocacy, Shared Responsibility, and Gendered Cultural Barriers; (5) Education and Community Influence as Transformative Levers; (6) Structural and Health System Inequities; and (7) Social and Institutional Linkages for Change. Together, these themes highlight how personal experiences, relationships, cultural expectations, and systemic factors shape engagement with PCC and provide the framework for the findings presented cohesively below (**S3 File**).

### 1. Fragmented knowledge pathways and practices across reactive, guided, and constrained routes

Knowledge about PCC was fragmented and largely reactive, shaped by past experiences and health events. Across hospitals, preparation for pregnancy often occurred only after complications or repeated pregnancy losses. This pattern was common among women with prior adverse reproductive experiences: "*I had several pregnancy losses before, and because of that, I started visiting the hospital more often… I decided to follow their advice and started the follow-up to prepare my body better before trying again*" (P06-F). Similarly, many first-time parents only recognized the need for

**Table 1. Participants' characteristics (N = 18).**

| Characteristic | Total (N = 18) |
|---|---|
| **Gender** | |
| Women | 10 |
| Men | 8 |
| Age (years) | |
| 19–24 | 5 |
| 25–34 | 4 |
| 35–45 | 9 |
| **Marital status** | |
| Married | 8 |
| In a relationship (not married) | 5 |
| Divorced/Single/Unmarried | 5 |
| **Education** | |
| Primary/Secondary (≤10th) | 6 |
| College/University | 8 |
| Postgraduate | 4 |
| Employment status | |
| Employed | 6 |
| Student | 4 |
| Unemployed/Casual/Street work | 5 |
| Self-employed (small business) | 1 |
| **Reproductive status** | |
| Pregnant | 5 |
| Partner currently pregnant | 2 |
| Planning to conceive | 5 |
| Recent pregnancy history | 3 |
| Fertility issues | 1 |
| Accompanied by infant care | 1 |

Note. Participants were recruited from two hospitals equally

preconception guidance after discovering pregnancy. Men generally became aware of PCC indirectly, often through crises such as infertility. This reactive pattern is not incidental; it reflects a deeper structural mechanism in which PCC is institutionally positioned as a response to reproductive failure rather than a preventive standard of care. When health systems do not proactively offer or visibly promote PCC, individuals default to crisis-driven engagement, normalizing reactive help-seeking as the only available pathway.

Social position, marital status, and age did not operate as independent barriers but converged to produce compounded disadvantages. Unmarried and young women faced stigma and exclusion that closed off formal pathways to PCC knowledge, while economic strain further narrowed available options, particularly among men and women with lower incomes. This triad of gender, age, and socioeconomic position functioned as an interlocking mechanism: each dimension amplified the constraining effects of the others, such that the most socially marginalized individuals faced the greatest informational and service deficits. Conversely, married and wealthier women accessed information and services more readily, not simply because of individual advantage, but because the health system itself was structurally more responsive to their social position, embedding inequity into routine service delivery.

Sources and pathways of knowledge were piecemeal, informal, and unevenly distributed, a pattern that cannot be understood apart from the systemic gaps that produced it. Where formal health services failed to provide proactive, accessible PCC guidance, participants turned to pharmacies, social networks, and digital media, not by preference, but by necessity. This informal turn itself generated further inequity: quality and completeness of information varied considerably across these channels, and gendered learning pathways emerged as men disproportionately relied on social media while women navigated pharmacy and peer networks. These divergent pathways reinforced rather than resolved gendered knowledge gaps. Faith shaped interpretations by legitimizing care for some while reinforcing fatalism for others, operating as a cultural mediator that either enabled or constrained engagement depending on how religious frameworks intersected with individual circumstances. Across participants, readiness was recognized as multidimensional, encompassing emotional and relational preparation alongside biomedical measures: *"It's not just pills. You need to be okay in your heart and life"* (P09-F), a recognition that participants themselves had arrived at despite, rather than because of, the care they received.

Practices of PCC were uneven, and this unevenness was not simply a reflection of individual motivation but of the interaction between resources, social support, and systemic structures. Women with supportive partners, family, and financial means were more likely to engage in dietary changes, health checks, or supplement use, not because they were inherently more health-conscious, but because the conditions enabling engagement were structurally available to them. Men's indirect contribution through finances or transport, while meaningful, rarely extended to clinical encounters, a pattern produced by the intersection of gender norms and health service design: services were not structured to include or invite men, and cultural expectations did not position male clinical engagement as normative. Unmarried women and those lacking financial or social support were consequently pushed toward informal channels, where unsafe self-medication became a predictable outcome of systemic exclusion rather than individual choice. Structural barriers, staff shortages, overcrowding, weak referral networks, and domestic workload did not operate in isolation but compounded one another, creating an environment in which sustained PCC engagement was structurally improbable for the most vulnerable. Yet despite these constraints, participants derived emotional and relational meaning from preconception preparation, with reassurance, calm, and pride in preventive action emerging as common threads, suggesting that the demand for PCC exists and is meaningful, but that structural conditions systematically fail to meet it.

## 2. Preventive orientation and reassurance

Participants displayed a strong preventive orientation toward PCC, seeing it as protective and beneficial across generations. Women consistently framed it as preparation to ensure healthier offspring and broader societal benefit: *"If we take care early, we give birth to children who can help our country tomorrow"* (P06-F). This framing is significant as it reveals how women have internalized PCC as a moral and civic responsibility, a dynamic that reflects broader cultural scripts positioning women as primary guardians of reproductive and family health. Men recognized indirect benefits, emphasizing the calming effect on households and the stability brought by joint engagement, but this difference in framing is not merely attitudinal. It reflects a gendered division of reproductive labor in which women bear the weight of preparation while men occupy a facilitative and peripheral role, a division that is simultaneously produced and reinforced by health service structures that direct PCC almost exclusively at women.

Responsibility for PCC was understood as collective but filtered through social roles and gender norms in ways that reproduced rather than challenged existing inequities. Women positioned themselves as primary actors, often actively encouraging men's involvement, while men highlighted financial, logistical, or supportive roles. This different was not simply a matter of individual preference but the product of intersecting forces: gender norms, institutional practices, and service design converged to construct PCC as women's domain. Field observations confirmed this mechanism directly, ANC talks about diet, folic acid, and timing were directed exclusively at pregnant women, structurally rendering PCC invisible to men and actively excluding those seeking pre-pregnancy guidance. Younger men demonstrated greater willingness

to engage in clinical visits, while older men expressed hesitancy, suggesting that generational shifts in gender norms are underway but that institutional practices have not adapted to enable or sustain this emerging engagement. Access to preventive knowledge and reassurance was uneven, shaped by the interaction between health system practices and social context rather than by individual motivation alone. Married women frequently felt entitled to services, while unmarried women, though excluded by clinics, emphasized early preparation to avoid later harm, navigating the contradiction between their exclusion and their preventive intent. This illustrates a critical mechanism: institutional gatekeeping based on marital status does not eliminate demand for PCC but displaces it into informal, less safe channels. Faith further mediated preventive practices, functioning not as a uniform influence but as a variable lens through which participants reconciled medical guidance with spiritual frameworks, enabling engagement for some while reinforcing fatalism for others, depending on how religious and biomedical narratives were locally interpreted and negotiated.

PCC was tied to responsible parenthood, moral duty, and collective well-being, yet this strong normative endorsement coexisted with significant emotional burden, a tension that reveals the gap between cultural valuation of PCC and the structural conditions necessary to support it. Even when preventive orientation was strong, feelings of fear, blame, and guilt emerged, reflecting the complex interplay of individual motivation, gendered expectations, and societal norms. These emotional costs were not evenly distributed: they were highest among those whose social position, unmarried, young, low-income, placed them furthest outside the boundaries of institutionally sanctioned care, suggesting that emotional burden functions as both a consequence and a mechanism of structural exclusion.

### 3. Emotional struggles and coping mechanisms

Participants framed PCC as deeply intertwined with emotional and psychological experience, and this entanglement cannot be understood as simply individual in origin. Pregnancy loss often served as a turning point, converting grief into motivation for preparation: *"Only after I lost the baby did I hear about pre-pregnancy care"* (P06-F). This pattern reveals a systemic failure with emotional consequences when PCC is not proactively offered; adverse reproductive outcomes become the primary entry point into care, meaning that emotional trauma and preventive engagement are structurally linked. Supportive encounters with health workers after complications helped rebuild confidence and re-engage women in preventive care, demonstrating that provider behavior operates as a critical lever: the same health system that produces emotional distress through structural neglect also has the capacity to mitigate it through relational care. Fear, stigma, and secrecy consistently constrained engagement with PCC, particularly for marginalized groups, but these emotional responses were not simply psychological reactions; they were socially produced. Young or unmarried women navigated care in silence to avoid judgment, and institutional exclusion further intensified feelings of isolation and vulnerability. Financial strain and lack of partner support heightened anxiety in ways that compounded rather than merely added to other barriers, illustrating how gender, marital status, and socioeconomic position interact to create layered emotional vulnerability in which each dimension intensifies the others. Hidden suffering among women whose partners withdrew support further reinforced that emotional risk is inseparable from relational and structural context.

Emotional support functioned as a critical coping mechanism, and its impact reveals the relational architecture underlying PCC engagement. When men provided reassurance or accompanied partners, fear and anxiety decreased: *"If the clinic welcomed us and said this is also for fathers, it would make a big difference" (P11-M).* This observation exposes a missed institutional opportunity; male inclusion in PCC not only addresses gendered exclusion but generates protective emotional effects for women, meaning that service redesign to include men would produce relational and psychological benefits beyond its direct gender equity value. Joint preparation strengthened trust and calm within couples, while women frequently associated PCC with hope and emotional reassurance, suggesting that the emotional dimensions of PCC are not peripheral but central to its uptake and sustainability.

Ambivalence and guilt persisted even among participants who valued preventive preparation, a pattern that illuminates the tension between normative endorsement of PCC and the social conditions that complicate its pursuit. Men described

concerns about ridicule for seeking preventive care, while women and men alike reported tension between responsible planning and internalized societal or faith-based expectations. These dynamics demonstrate that emotional struggles surrounding PCC are not individual failings but structurally and culturally produced responses to contradictory social demands, demands that simultaneously valorize reproductive responsibility and stigmatize the very behaviors through which that responsibility might be enacted.

### 4. Advocacy, shared responsibility, and gendered cultural barriers

Participants emphasized advocacy as essential to making PCC visible and routine. Schools, faith leaders, and elders were identified as critical entry points to reduce stigma and legitimize care. Integration of PCC into public maternal and child health services, combined with opportunistic counseling during routine visits and media-based messaging, was viewed as an effective strategy when communication was "*friendly and understandable*" (P15-M). Participants also highlighted that advocacy should actively include men and unmarried women to normalize preventive care across the population. Cultural beliefs, gender norms, and stigma strongly constrained both advocacy and access. Marriage emerged as a dominant gatekeeping mechanism, defining who was considered eligible for care. Unmarried and young women frequently faced exclusion, while gossip, ridicule, and social policing within communities reinforced silence and discouraged care-seeking. Faith played a dual role, legitimizing PCC for some while promoting fatalistic interpretations for others. These dynamics illustrate how intersectional factors, including age, marital status, faith, and social norms, interact across community and cultural levels to shape the legitimacy and acceptance of PCC.

Gendered expectations created unequal responsibilities and reinforced structural barriers. Women often carried the burden of preparation, whereas men's participation was commonly framed as financial or logistical support. Masculine norms and economic pressures discouraged clinic attendance and limited male involvement. Men's authority over reproductive decisions sometimes constrained women's autonomy, while social and familial blame disproportionately targeted women when pregnancy complications occurred. Economic and power imbalances further increased vulnerability, particularly for young women who depended on partners for support. Despite these structural and cultural barriers, participants recognized the need for shared responsibility. Families, communities, faith institutions, schools, and policymakers were seen as collectively accountable for creating an enabling environment for PCC. Weak systems, limited-service quality, and insufficient advocacy were perceived as reinforcing inequality. Participants suggested that stronger engagement of men, public endorsement by community leaders, and consistent community-level messaging could help normalize PCC and reduce stigma. These findings illustrate how gender, marital status, social norms, economic position, and institutional systems intersect across socio-ecological levels to influence PCC access and legitimacy.

### 5. Structural and health system inequities

Participants described PCC as constrained by financial hardship, fragmented services, and institutional neglect, but these constraints did not operate independently; they formed an interlocking system of exclusion in which each barrier amplified the others. For many, daily survival took priority over preventive care: "*Because in the city, there are many pressures, rent, transport, food costs. If you earn something small, you prioritize surviving, not going to the clinic… people may laugh and say, 'Why are you worrying like a woman?'*" (P14-M). This quote is analytically significant on multiple levels: it reveals not only how economic precarity displaces preventive health-seeking but also how gender stigma is woven into the very language of financial constraint, making male engagement in PCC socially costly as well as economically difficult. The perception of PCC as a privilege of the wealthy was therefore not simply a reflection of individual circumstance but a socially produced understanding shaped by the absence of visible, free, and routinely offered preventive services, a structural condition that communicates to low-income populations that PCC is not intended for them.

Inconsistent service delivery did not merely inconvenience participants; it actively eroded the trust and motivation necessary to sustain preventive engagement. Long waits, absent providers, staff shortages, and dismissive attitudes

functioned as cumulative deterrents: *"You go today, and the provider is not there, so you return home" (P06-F)*. Each failed encounter reinforced the perception that the health system does not prioritize PCC, making subsequent help-seeking less likely. The resulting drift toward pharmacies as a default entry point for reproductive health needs was therefore not a free choice but a predictable outcome of institutional failure, one that simultaneously exposed participants to the risks of limited counseling and self-medication while removing them further from the formal care pathway. The affordability of private clinics relative to their perceived respectfulness further illustrates how the quality of care and the dignity of the care experience are themselves distributed unequally along economic lines, embedding inequity into the texture of the clinical encounter.

Systemic gaps at the programmatic and policy level revealed a structural logic in which PCC was rendered invisible through its exclusion from mainstream service delivery. The failure to integrate PCC into maternal and child health services meant that the most natural point of contact for reproductive-age individuals, existing health visits, did not function as an entry point into preventive care. Men's exclusion from services was not simply an oversight but a product of institutional design that reflected and reinforced prevailing gender norms, creating a self-sustaining cycle: services were not designed for men because men were not expected to attend, and men did not attend because services were not designed for them. Youth-focused programs that prioritized HIV or contraception over broader reproductive preparedness similarly reflected policy assumptions that inadvertently narrowed the scope of preventive reproductive health for young people, leaving pre-pregnancy preparation unaddressed at the precise life stage when it could be most impactful.

## 6. Social and institutional linkages for change

Participants emphasized that PCC could only be sustained through strong family, community, and institutional linkages, and the mechanisms through which these linkages operated reveal how social context either enables or forecloses preventive engagement. Families were identified as the first circle of influence, where even small acts of support reinforced dignity and readiness: *"My sister-in-law once wanted to check her blood. She called her husband from the clinic, and he sent transport money right away. She felt respected and continued her visits"* (P06-F). This account illuminates a critical mechanism; partner support did not merely remove a logistical barrier but transformed the emotional experience of care-seeking, converting a potentially stigmatizing encounter into one of relational affirmation. The capacity to continue preventive visits was therefore contingent not only on financial access but on the relational conditions that make care-seeking feel safe and dignified.

Community endorsement operated as a norm-reinforcing mechanism that extended beyond individual households to reshape collective expectations. Leaders and elders were identified as gatekeepers whose visible endorsement could neutralize the social risk of seeking PCC, particularly for unmarried women and men whose engagement transgressed prevailing gender expectations. Men's visible participation was understood not merely as individual support but as a social signal that modeled and legitimized broader norm change: *"When others see a husband involved, it gives them courage too"* (P16-M). This reveals how behavioral change in PCC is socially contagious; individual acts of engagement, when publicly visible, create permission structures that lower the threshold for others. The mechanism here is not persuasion but social demonstration, suggesting that community-level interventions focused on visibility and modeling may be more effective than those relying solely on information provision.

Institutional continuity was equally vital, and its absence exposed a structural contradiction at the heart of PCC delivery. Health extension workers played a key role in introducing PCC, yet weak linkages between hospitals, frequent stock-outs, and inconsistent service delivery undermined the trust they helped to build, effectively neutralizing at the system level the relational gains achieved at the community level. This dynamic illustrates how macro-level institutional failures can override micro-level relational successes, highlighting that community engagement strategies will remain insufficient without corresponding investment in service reliability and supply chain integrity. Participants' calls for accessible check-ups, youth-friendly spaces, respectful clinical encounters, and stronger PCC integration within existing health systems were not

simply wish-lists but articulations of the specific conditions under which sustained preventive engagement becomes structurally possible, and their absence explains, more precisely than individual barriers alone, why PCC remains inaccessible to those who need it most.

### 7.  Social and institutional linkages for change

Participants emphasized that PCC could only be sustained through strong family, community, and institutional linkages. Families were identified as the first circle of influence, where even small acts of support reinforced dignity and readiness. "*My sister-in-law once wanted to check her blood. She called her husband from the clinic, and he sent transport money right away. She felt respected and continued her visits*" (P06-F). Conversely, unmarried women experienced vulnerability due to social stigma and lack of visible pregnancy. Community endorsement was framed as reinforcing household effort**s**. Leaders and elders were seen as gatekeepers capable of reducing gossip and legitimizing care, while men's visible participation inspired broader norm change: "When others see a husband involved, it gives them courage too" (P16-M). Such social mechanisms were considered pivotal in overcoming intersectional barriers related to gender, marital status, and age, creating collective accountability and modeling engagement within communities.

Institutional continuity was equally vital for maintaining PCC. Health extension workers played a key role in introducing PCC, yet weak linkages between hospitals, frequent stock-outs of supplies, and inconsistent service delivery undermined trust and continuity of care. Participants highlighted that these disruptions often discouraged individuals from continuing preventive efforts. Participants emphasized that sustainability depends on government commitment and tangible action. Beyond advocacy alone, visible and reliable services were seen as crucial. Calls were made for accessible check-ups, youth-friendly service spaces, respectful care in clinics, and stronger integration of PCC within existing health systems. Preventive benefits were also framed collectively, linking early preparation for pregnancy to healthier families and broader societal well-being.

## Discussion

This study extends understandings of PCC by showing it is not merely a biomedical checklist but a socially embedded, relational, and contested practice within the urban selected hospital context of Addis Ababa. Participants' accounts emphasized that PCC knowledge and practices were filtered through stigma, gender roles, faith, economic constraints, and structural neglect. The findings illustrate how PCC is foreclosed not only at the level of individual awareness but also within interpersonal relationships, community expectations, and systemic arrangements. This echoes international work showing PCC as a hidden and fragile concept, poorly integrated into public health promotion [31], and reinforces the call for more inclusive, equity-oriented definitions of PCC [32]. It also aligns evidence from Sub-Saharan Africa highlighting substantial gaps in awareness, risk identification, and systematic implementation of preconception care services across health systems [33].

A key insight is the fragility of knowledge. While participants recognized certain components such as nutrition, folic acid, and rest, this knowledge was fragmented, inconsistently delivered, and often acquired too late, frequently after miscarriage or pregnancy complications. This resonates with research from Nepal and Ghana that attributes late PCC uptake to systemic gaps rather than ignorance [34,35] as well as with Australian findings of piecemeal, reactive learning [31]. In this study, fragmentation was not only informational but also structural: access to knowledge was routinely mediated by stigma, marital status, and gender norms. This reframes the problem away from "low awareness" toward what Dean et al. [23] termed the structural invisibility of PCC within public health systems, a theme the concept analysis [32] further advances by emphasizing systemic and intergenerational determinants. Similar patterns of fragmented knowledge and delayed engagement have also been reported in Ethiopian contexts, where awareness of PCC is often limited and information is inconsistently communicated within health services [36].

Beyond informational gaps, participants described PCC as a form of relational readiness. Women emphasized emotional stability and dignity, men stressed financial preparedness, and couples highlighted trust and reduced conflict when planning together. These findings align with Waggoner and Pentecost [37] critique of PCC as a relational practice that cannot be reduced to technical interventions, while also extending it by grounding readiness in everyday negotiations of stigma and legitimacy. The emphasis on reassurance, household calm, and social dignity parallels the Australian study's insight that PCC was understood less as a biomedical act than as a preparation for future roles and identities [27].

The role of men presented both challenges and opportunities. While men rarely perceived PCC as directly relevant to their own health, they framed themselves as facilitators of women's care. This echoes global patterns of men's exclusion in reproductive services [5], but the accounts here extend the discussion by revealing men's latent desire for inclusion. Their reluctance stemmed less from disinterest than from the absence of men-friendly entry points and the stigma of being seen as "worrying like a woman." This illustrates how hegemonic masculinity intersects with institutional neglect to render men invisible within PCC, even as their involvement is critical for women's access. Studies found similar uncertainty among men, who recognized the importance of preconception health but struggled to locate their own role [27]. Programs should therefore engage men not only as supporters but also as clients, with tailored services addressing fertility, STIs, and emotional well-being, an approach consistent with the inclusivity emphasized in the concept analysis [28].

Cultural norms and stigma acted as decisive gatekeepers of legitimacy, reinforcing the privatization of PCC. Marriage was consistently described as the threshold for access, gossip deterred unmarried women, and clinic visits carried the risk of moral judgment. While some women resisted these pressures through secrecy or selective disclosure, others internalized fatalism: "*if God gives, you will have a child.*" Intersectional analysis reveals how age, gender, marital status, and poverty combined to amplify vulnerability: Young women faced peer surveillance, unmarried women confronted institutional denial, and men risked ridicule. This supports intersectionality theory's insight that disadvantages are not additive but mutually reinforcing, shaping who can access care and under what conditions. These findings build on, but also complicate, a study that identified stigma and silence as barriers but did not encounter the same intensity of marital gatekeeping or community surveillance, suggesting contextual variations in how legitimacy is constructed. Comparable qualitative evidence from Northern Ethiopia similarly reports that cultural expectations, social stigma, and limited awareness strongly shape how individuals perceive and access PCC services [36].

Yet participants were not passive. Improvised practices, using pharmacies discreetly, relying on social media, or disguising healthcare visits, demonstrated everyday agency in contexts of exclusion. These strategies complicate narratives of "*low demand,*" revealing instead how individuals actively negotiated stigma and structural service gaps. Such hidden practices resonate with Scott's concept of everyday resistance, in which subordinate groups develop informal, often invisible strategies to navigate and survive within restrictive structures, where open defiance is neither safe nor possible [38]. They also resonate with global accounts of "*workarounds*" in reproductive health, where women and couples create informal care pathways when systems fail to meet their needs [27].

The findings also underscore the importance of language and communication. Participants frequently critiqued the term "PCC" as technical, elitist, or women-centered, advocating instead for simpler local terms such as "health before pregnancy." This highlights how linguistic framing can itself be a barrier to uptake, shaping perceptions of exclusivity and limiting resonance in community contexts. The concept analysis [28] explicitly calls for clarity in definitions, noting how inconsistent terminology fragments both practice and research. Our findings add empirical weight, showing how terminology is not only an academic issue but also a lived barrier to acceptance.

Health system barriers magnified inequities. Public hospitals lacked formal PCC, advice was incidental, and providers were often absent or unresponsive. As in studies from Uganda and Ethiopia, PCC was perceived as available only in private facilities, reinforcing its status as a privilege for the wealthy. Economic barriers, overcrowding, and negative provider attitudes further deterred engagement. These findings point to the need for systemic integration, as emphasized by WHO [7], which calls for PCC to be embedded within routine maternal and adolescent health services and supported

by cross-sectoral coordination. These findings show that systemic neglect not only limits PCC access but deepens stigma and exclusion, echoing previous studies [27,31,32] on its invisibility in health promotion while also aligning with evidence from Sub-Saharan Africa that highlights persistent preconception health risks and limited integration of PCC services within health systems [36]. At the same time, participants identified clear pathways for transformation. They envisioned PCC as a collective responsibility spanning families, peers, schools, faith leaders, providers, and the state. Suggested strategies, embedding PCC in schools, using mass media, engaging men and role-model couples, and employing respectful provider communication, mirror the socio-ecological model's emphasis on multi-level interventions. This aligns with global recommendations for life-course and relational approaches, but this study adds granularity by showing how these strategies resonate within Ethiopian realities of stigma, economic strain, and institutional neglect.

## Strengths and limitations

This study offers one of the first in-depth interpretive descriptive qualitative analyses of PCC using socio-ecological and intersectional lenses to reveal gendered, cultural, and systemic dynamics. Its strength lies in the rigorous, context-rich interpretation that links lived experiences with structural gaps in PCC awareness, access, and delivery. However, the study was conducted in Addis Ababa, Ethiopia's largest and most well-resourced city, so participants' experiences may differ from those in smaller or peripheral urban areas. Accordingly, the transferability of the findings to other urban contexts should be considered with caution. Additionally, as with many qualitative studies conducted in multiple languages, minor meaning loss may have occurred during translation despite careful efforts to preserve participants' intended meanings.

## Implications

**Practice:** Providers should adopt relational, gender-sensitive approaches that address emotional readiness, stigma, and cultural norms alongside biomedical risk reduction. Men might be engaged explicitly as both clients and supporters, while adolescents and unmarried women require confidential, stigma-free entry points. Services should be visibly accessible, delivered in plain, locally grounded language, and supported by trusted community figures to normalize preventive care.

**Policy:** Although national PCC guidelines exist, implementation remains limited. Strengthening integration requires clear operational guidance, inclusive eligibility criteria, and consistent service delivery. Community-driven approaches that leverage schools, peer networks, digital platforms, and faith leaders should complement formal services to increase uptake and normalize PCC.

**Research:** Future studies should move beyond clinical outcomes to explore how language, stigma, gender norms, and systemic neglect shape PCC engagement. Interventions that are men-inclusive, community-driven, and digitally mediated should be tested to assess effectiveness across intersectional contexts.

**Health Systems:** Equitable access depends on reliable referral networks, workforce support, reduction of indirect costs, and continuous service availability. Without institutional commitment, PCC remains fragmented, reactive, and accessible primarily to those with resources.

## Conclusion

PCC is not merely a bundle of biomedical interventions but a relational and socio-culturally-embedded practice. The findings show that fragmented knowledge pathways, emotional challenges, gendered expectations, and structural health-system gaps shape how individuals access and experience PCC. Improving PCC implementation, therefore, requires coordinated action at multiple levels.

At the clinical and service delivery level, healthcare providers should integrate PCC into routine reproductive health consultations, including family planning and antenatal care visits, using structured counseling protocols aligned with the 2024 national PCC guideline. Gender-sensitive service models, such as couple-friendly consultation spaces and

male-inclusive appointment pathways, should be piloted in primary healthcare facilities to address documented patterns of male exclusion. Emotionally supportive and confidential care approaches should be embedded into provider training modules, with competency-based assessments to ensure consistent implementation. At the health system level, operationalizing the 2024 national PCC guideline should include facility-level service redesign, such as designated PCC consultation times, standardized referral pathways, and integration of PCC into community health extension worker training curricula. Regulatory mechanisms for pharmacy-based reproductive health advice and digital health platforms should be strengthened to ensure information quality and safety. At the community level, faith leaders, community health workers, and educational institutions should be engaged through structured awareness programs that promote accurate PCC information, address stigma, and foster shared reproductive responsibility among men, women, and families. These strategies can support the transition of PCC from a reactive and privilege-dependent practice to a visible, accessible, and preventive standard of care.

## Supporting information

**S1 File. Semi-structured interview guide for reproductive-age individuals.**
(DOCX)

**S2 File. COREQ (Consolidated Criteria for Reporting Qualitative Research) checklist.**
(PDF)

**S3 File. Illustrative participant quotations supporting thematic findings.**
(DOCX)

## Author contributions

**Conceptualization:** Yared Asmare Aynalem, Pauline Paul, Salima Meherali.

**Data curation:** Yared Asmare Aynalem.

**Formal analysis:** Yared Asmare Aynalem.

**Funding acquisition:** Yared Asmare Aynalem.

**Investigation:** Yared Asmare Aynalem.

**Methodology:** Yared Asmare Aynalem, Pauline Paul, Zohra S. Lassi, Salima Meherali.

**Software:** Yared Asmare Aynalem.

**Supervision:** Pauline Paul, Zohra S. Lassi, Salima Meherali.

**Validation:** Yared Asmare Aynalem.

**Visualization:** Yared Asmare Aynalem.

**Writing – original draft:** Yared Asmare Aynalem.

**Writing – review & editing:** Pauline Paul, Zohra S. Lassi, Salima Meherali.

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
