## [Decision Letter · Decision Letter 0]

17 Feb 2026

PONE-D-25-65367Understanding Reproductive-Age Individuals’ Perspectives on Preconception Care in Ethiopia: Social, Cultural, and Structural DeterminantsPLOS One

Dear Dr. Aynalem,

Thank you for submitting your manuscript to PLOS ONE. After careful consideration, we feel that it has merit but does not fully meet PLOS ONE’s publication criteria as it currently stands. Therefore, we invite you to submit a revised version of the manuscript that addresses the points raised during the review process.

We look forward to receiving your revised manuscript.

Kind regards,

Maher Abdelraheim Titi

Academic Editor

PLOS One

Journal Requirements:

“We acknowledge the participants and the administrators of Gandhi Memorial Hospital and Zewditu Memorial Hospital for their collaboration. We also acknowledge funding support from the Vanier Canada Graduate Scholarship and the International Development Research Centre (IDRC) Doctoral Research Award. Institutional support from Debre Berhan University and the University of Alberta is gratefully recognized.”

“YAA received the Vanier Canada Graduate Scholarship (Government of Canada; https://vanier.gc.ca

) and the IDRC Doctoral Research Award (International Development Research Centre; https://idrc-crdi.ca

). The funders had no role in study design, data collection and analysis, decision to publish, or preparation of the manuscript”

“NO authors have competing interests”

6. We note that you have indicated that there are restrictions to data sharing for this study. PLOS only allows data to be available upon request if there are legal or ethical restrictions on sharing data publicly. For more information on unacceptable data access restrictions, please see http://journals.plos.org/plosone/s/data-availability#loc-unacceptable-data-access-restrictions.

Additional Editor Comments :

The manuscript is rich and well‑constructed, employing an appropriate and well‑justified qualitative methodology to explore preconception care (PCC) in urban Ethiopia.

• Page 16(Study Setting): The authors should briefly clarify the rationale for selecting only two hospitals in Addis Ababa and how these sites were expected to provide diverse perspectives.

• The study was conducted exclusively in Addis Ababa, Ethiopia’s largest and most well resourced city. Individuals in smaller or peripheral urban areas may have different experiences with PCC due to variations in resources, cultural norms, and health system access. Please acknowledge this in the Limitations section and clarify that the findings should not be generalized to all urban Ethiopian populations.

• Page 17 (Data Collection): The manuscript states that: “Drafts were pilot tested for clarity and cultural appropriateness before use.” it is unclear how the pilot testing was conducted, with whom, whether pilot interviews were included in the final analysis and what specific modifications were made as a result. Further elaboration is recommended.

• Page 19: The authors state that information power guided data collection, which is appropriate for Interpretive Description. However, the manuscript does not explain how sufficient information power was determined. Which dimensions and what indicators were considered?

• Page 21 (Overview of Findings): The results section is rich and insightful; however, it is excessively detailed, with long narrative passages that may overwhelm readers. To improve readability, consider shorten narrative descriptions and moving long quotations to supplementary materials to improve readability.

• Page 21 (Discussion): The Discussion is strong but at times repeats results rather than synthesizing them. Some paragraphs are long and dense. Please reduce repetition of participant quotes and focus on interpretation rather than restatement.

• Some paragraphs are overly long and would benefit from being broken into shorter, more focused sections to improve readability.

• Numbering and page numbering are needed throughout the manuscript to support the review process.

Reviewers' comments:

Reviewer's Responses to Questions

**Comments to the Author**

1. Is the manuscript technically sound, and do the data support the conclusions?

Reviewer #1: Yes

Reviewer #2: Yes

2. Has the statistical analysis been performed appropriately and rigorously? 

Reviewer #1: N/A

Reviewer #2: Yes

3. Have the authors made all data underlying the findings in their manuscript fully available?

Reviewer #1: No

Reviewer #2: Yes

4. Is the manuscript presented in an intelligible fashion and written in standard English?

Reviewer #1: Yes

Reviewer #2: No

5. Review Comments to the Author

Reviewer #1: Reviewer’s comment to the Author

PLOS One

Understanding Reproductive-Age Individuals’ Perspectives on Preconception Care in Ethiopia: Social, Cultural, and Structural Determinants--Manuscript Draft- Manuscript Number: PONE-D-25-65367

Thank you for giving me a chance to review this qualitative research on preconception care.

General comments

1. The manuscript addresses important issues that could contribute to / have implications for healthcare services in Ethiopia. The manuscript is well written overall. The content, introduction, and discussion sections are clearly and effectively articulated. However, the Results section requires substantial revision and reanalysis. The themes and subthemes should be more clearly derived from / aligned with the specific research question(s).

2. The document does not include line numbers, which would be helpful for reviewers to refer to specific lines precisely.

3. Please provide/attach the interview guide (or topic guide) used in the study. This is important for assessing how the emerged categories and subcategories in the results relate to the questions asked.

4. Please ensure the manuscript matches the formatting guidelines of PLOS ONE

Title:

5. I suggested the title should be modified as…” Exploring Knowledge, Attitudes, and Practices Regarding Preconception Care Among Reproductive-Age Individuals in Ethiopia: An Interpretive Description Study Using the Socio-Ecological and Intersectional Models”

Abstract:

6. You said, in Ethiopia, PCC remains almost absent in everyday health practice; the pooled utilization of PCC in Ethiopia is 26%. “The magnitude of preconception care utilization and associated factors among women in Ethiopia: systematic review and meta-analysis, 2024 | BMC Pregnancy and Childbirth”, can we say almost absent? Please justify.

7. Similarly, you mentioned again: “This study provides the first in-depth, urban Ethiopian analysis showing how adults actively negotiate preconception care (PCC) within intersecting gender, moral, and structural systems.” However, more than three published qualitative studies on preconception care already exist in Ethiopia. As a researcher, particularly a qualitative one, honesty is essential to maintain the trustworthiness and credibility of the study.

8. You employed maximum variation sampling, yet your study participants were drawn from the reproductive-age group (both women and men), which constitutes a relatively homogeneous population. Do you believe maximum variation sampling is appropriate for this study?

9. Is interpretive description design more dominant in exploratory studies

Or Explanatory studies? Make sure that you use the appropriate design for your research.

10. Are all the listed keywords scientifically appropriate and aligned with standard terminology? Please modify the words included in the keywords.

Introduction:

11. What is the primary objective of this study—to explore the knowledge, attitudes, and practices (KAP) regarding PCC, or to examine participants’ perspectives on PCC within socio-ecological and intersectional contexts? Please justify your response by referring to the study title and conclusion, as there appears to be a potential mismatch between the stated aim (KAP-focused) and the interpretive, contextual emphasis in the title and conclusion.

Methods

12. Please attach the written informed consent document.

13. The guides explored knowledge, attitudes, and practices regarding PCC; perceived benefits and barriers; gender and cultural dynamics; and experiences with healthcare providers.

Result

14. You mentioned that pilot tests were conducted for clarity and cultural appropriateness before use. Could you elaborate on the number of participants included in the pilot, and was this pilot data included in the formal analysis? Please describe this process in detail within the data collection section.

15. Five women who were planning to conceive were included. How were these women identified? What method or tool was used for their identification?

16. Detailed descriptions within each theme and subtheme are vital for rigorous analysis and reporting.

17. The sub-themes within the seven main themes should be rewritten to consolidate similar concepts.

18. Since the main purpose of using maximum variation sampling is to collect information from a heterogeneous participant group, the analysis should describe and compare the differences and similarities among participants within each theme and subtheme.

19. Please specify which conceptual framework or model (e.g., the socio-ecological model) guided this study. The results section should then discuss the main findings in relation to this chosen framework.

20. The overall results presentation seems shallow, incoherent, and vague. The data likely requires re-analysis, and the presentation of themes and subthemes needs to be updated accordingly.

Discussion

21. Is the word "must" commonly used to formulate recommendations for different actors based on the findings of a single study?

22. You state, "Policy: Ethiopia’s reproductive health policy should integrate PCC into public services with clear guidelines." However, a national preconception care guideline was developed in 2024. What is the intended implication of your recommendation, given this existing policy?

Conclusion

23. In your conclusion, you stated, “PCC should combine biomedical prevention with relational and gender-sensitive practice.” This statement is unclear. According to the Ethiopian PCC guideline, the definition of PCC already includes a bundle of interventions aimed at identifying and modifying biomedical, behavioral, and social risks to improve pregnancy outcomes. Why, then, does your conclusion suggest that PCC should combine biomedical prevention? The guideline already encompasses this approach.

24. Similarly, in your conclusion, you wrote, “Providers must engage men and women as clients and supporters while ensuring confidential, stigma-free access for adolescents and unmarried women.” According to the Ethiopian PCC guideline, couple-based services are included as a guiding principle. Your statement should more directly align with this principle of couple-centered care.

25. Therefore, your implications and conclusions should be updated to reflect the context of the Ethiopian national PCC guideline.

Reviewer #2: 1. Title Clarity and Focus

• Feedback: The title effectively captures the essence of the study, but consider making it more concise by removing less critical phrases. For example, "Understanding Reproductive-Age Individuals' Perspectives on Preconception Care in Ethiopia" could be shortened to "Perspectives on Preconception Care in Ethiopia: Social, Cultural, and Structural Determinants." This focuses the reader's attention on the primary research topic.

2. Establishing Context

• Feedback: The background provides essential context, particularly regarding the introduction of the PCC guidelines. However, consider including a brief mention of the specific health outcomes or public health significance associated with improved PCC in Ethiopia. This might enhance the urgency and importance of the study.

3. Methodological Rigor

• Feedback: The methods section outlines an appropriate approach for this type of qualitative research. It may be beneficial to elaborate on the selection criteria for participants and the rationale for employing maximum-variation sampling. This detail could enhance the robustness of your methodology and reassure readers about the representativeness of the sample.

4. Thematic Insights

• Feedback: The results section highlights significant themes that demonstrate the complexity of PCC uptake. To strengthen this section, consider providing brief examples or quotes from participants to illustrate the themes, particularly regarding "marital gatekeeping" and "emotional negotiations.” Rich qualitative data enhances the credibility and relatability of the findings.

5. Actionable Conclusions

• Feedback: The conclusions successfully address the complexities of PCC practice in Ethiopia. To further improve, consider specifying actionable recommendations based on your findings, such as potential interventions for healthcare providers, community organizations, or policymakers. Explicitly connecting these recommendations to identified themes could provide a clearer pathway for implementing the PCC guidelines effectively.

Overall Impression

Your work presents valuable insights into preconception care in Ethiopia, addressing gaps in the current literature by focusing on social and cultural determinants. By refining the title and enhancing specific sections, you can communicate your findings more effectively and emphasize the need for comprehensive, context-aware strategies in PCC implementation.

• What specific social and cultural factors significantly influence the acceptance and uptake of preconception care among different demographic groups in urban Ethiopia?

• How do gender norms and roles affect men’s involvement in preconception care, and what strategies can be implemented to encourage their active participation in this process?

• What barriers do married women face in accessing healthcare related to preconception care, and how can healthcare services be adapted to better meet their needs?

6. PLOS authors have the option to publish the peer review history of their article (what does this mean?). If published, this will include your full peer review and any attached files.

Reviewer #1: **Yes:** Gebremedhin Gebreegziabher Gebretsadik

Reviewer #2: No

---

## [Author Response · Author response to Decision Letter 1]

19 Mar 2026

Authors Response

Editor and Reviewer Comments Comments/ Revisions Made

Editor comment

Journal Requirements

Paper provides good context/background for Please ensure that your manuscript meets PLOS ONE's style requirements, including those for file naming. The PLOS ONE style templates can be found at

and

Response :We thank the Editor for this reminder. We have revised the manuscript to ensure full compliance with PLOS ONE style and formatting requirements, including correct file naming, and have used the provided templates.

Thank you for stating the following in the Acknowledgments Section of your manuscript:

“We acknowledge the participants and the administrators of Gandhi Memorial Hospital and Zewditu Memorial Hospital for their collaboration. We also acknowledge funding support from the Vanier Canada Graduate Scholarship and the International Development Research Centre (IDRC) Doctoral Research Award. Institutional support from Debre Berhan University and the University of Alberta is gratefully recognized.”

Response : Thank you for this clarification. The support received was in the form of a scholarship and award, and no specific funding number is associated with it. The funding information has been reported accordingly in the Funding Statement, and it has been removed from the Acknowledgments section in line with PLOS guidelines. We have also deleted in the submission portal.

“YAA received the Vanier Canada Graduate Scholarship (Government of Canada; https://vanier.gc.ca

) and the IDRC Doctoral Research Award (International Development Research Centre; https://idrc-crdi.ca

). The funders had no role in study design, data collection and analysis, decision to publish, or preparation of the manuscript”

Response :We thank the Editor for this clarification. We have removed all funding-related text from the Acknowledgments section. The corrected Funding Statement is provided in the cover letter, and we kindly request that the online submission form be updated accordingly.

We note that the grant information you provided in the ‘Funding Information’ and ‘Financial Disclosure’ sections do not match.

Response :Thank you for this comment. The support received for this study was in the form of a scholarship and an award, and no grant numbers were assigned. We have revised the funding information to ensure consistency between the Funding Information and Financial Disclosure sections.

Thank you for stating the following in your Competing Interests section:

“NO authors have competing interests”

Response :We thank you for this clarification. We confirm that the correct statement is: “The authors have declared that no competing interests exist.” This has been included in the cover letter, and we kindly request that the online submission form be updated accordingly.

PLOS requires an ORCID iD for the corresponding author in Editorial Manager on papers submitted after December 6th, 2016. Please ensure that you have an ORCID iD and that it is validated in Editorial Manager.

Response :Thank you for the reminder. The corresponding author’s ORCID iD has been provided and validated in Editorial Manager.

We note that you have indicated that there are restrictions to data sharing for this study. PLOS only allows data to be available upon request if there are legal or ethical restrictions on sharing data publicly.

a) If there are ethical or legal restrictions on sharing a de-identified data set, please explain them in detail.

b) If there are no restrictions, please upload the minimal anonymized data set necessary to replicate your study findings.

Response :Thank you for the comment. All relevant data are within the paper and its Supporting Information file.

Please include captions for your Supporting Information files at the end of your manuscript, and update any in-text citations to match accordingly. Thank you for the reminder. Supporting Information captions and in-text citations have been updated. See at the end of the document in the revised file.

If the reviewer comments include a recommendation to cite specific previously published works, please review and evaluate these publications to determine whether they are relevant and should be cited.

Response :Thank you for this comment. We have carefully reviewed the suggested publications and cited those that are relevant to the scope and context of our study.

Additional Editor Comments

Page 16 (Study Setting)

The authors should briefly clarify the rationale for selecting only two hospitals in Addis Ababa and how these sites were expected to provide diverse perspectives.

Response :Thank you for this comment. We have clarified in the Methods section the rationale for selecting the two hospitals and how they were expected to capture diverse perspectives.

Study conducted only in Addis Ababa

The study was conducted exclusively in Addis Ababa, Ethiopia’s largest and most well resourced city. Individuals in smaller or peripheral urban areas may have different experiences with PCC due to variations in resources, cultural norms, and health system access. Thank you for this important comment. The Limitations section has been revised to note that, consistent with qualitative research, and as the study was conducted in Addis Ababa, transferability to other urban Ethiopian settings with different resources and contexts should be interpreted with caution.

Page 17 (Data Collection)

The manuscript states that: “Drafts were pilot tested for clarity and cultural appropriateness before use.” it is unclear how the pilot testing was conducted.

analysis.

Response :Thank you for the comment. The interview guides were pilot tested with three participants similar to the study population, who were not included in the final sample. Please see the Methods section for details.

Page 19

The authors state that information power guided data collection, which is appropriate for Interpretive Description. However, the manuscript does not explain how sufficient information power was determined.

Response :Thank you for the comment. Sufficient information power was assessed based on the study aim, sample specificity, the depth and richness of participants’ narratives, and the interpretive description analytic approach. Please see the revised Methods section for clarification.

Page 21 (Overview of Findings)

The results section is rich and insightful; however, it is excessively detailed.

Response : Thank you for this suggestion. We have shortened the narrative and moved some longer quotations to Supporting Information to improve readability.

Discussion

The Discussion is strong but at times repeats results rather than synthesizing them.

Response :Thank you for this helpful comment. The Discussion has been revised to reduce repetition, shorten dense paragraphs, and strengthen interpretation rather than restating the results.

Paragraph length and numbering

Some paragraphs are overly long and would benefit from being broken into shorter sections.

Numbering and page numbering are needed.

Response :Thank you for this suggestion. We have revised the manuscript by breaking long paragraphs into shorter sections and adding continuous line and page numbering throughout..

Reviewer's Responses to Questions

1. Is the manuscript technically sound, and do the data support the conclusions?

Reviewer #1: Yes

Reviewer #2: Yes

Response :We thank the reviewers for confirming the technical rigor and that the data support the conclusions.

2. Has the statistical analysis been performed appropriately and rigorously?

Reviewer #1: N/A

Reviewer #2: Yes

Response :Thank you for your feedback.

3. Have the authors made all data underlying the findings in their manuscript fully available?

Reviewer #1: No

Reviewer #2: Yes . We have now fully complied with the PLOS Data policy. All underlying data have been provided as Supporting Information, and the Data Availability Statement has been updated accordingly.

Response :Thank you for the feedback

4. Is the manuscript presented in an intelligible fashion and written in standard English?

Reviewer #1: Yes

Reviewer #2: No The manuscript has been thoroughly revised to improve grammar, clarity, and overall readability.

Response :Thank you for your feedback.

Review 1 Comments to the Author

General comments

The manuscript addresses important issues that could contribute to / have implications for healthcare services in Ethiopia. The manuscript is well written overall. The content, introduction, and discussion sections are clearly and effectively articulated. However, the Results section requires substantial revision and reanalysis. The themes and subthemes should be more clearly derived from / aligned with the specific research question(s).

Response : Thank you for this constructive comment. The Results section has been revised to ensure that the themes and subthemes clearly align with the study's research questions. The coding framework and thematic structure were refined using an inductive interpretive description approach, and the findings are now presented through clearer themes and a more cohesive narrative, supported by representative quotations. Please see the color change in the revised document

The document does not include line numbers, which would be helpful for reviewers to refer to specific lines precisely.

Response :Thank you for noting this. Line numbers have now been added throughout the manuscript.

Please provide/attach the interview guide (or topic guide) used in the study. This is important for assessing how the emerged categories and subcategories in the results relate to the questions asked.

Response :Thank you for this comment. The interview guide has now been included as a supplementary file.

Please ensure the manuscript matches the formatting guidelines of PLOS ONE

Response :We thank the reviewer for this comment. We confirm that the manuscript and overall document are aligned with the PLOS ONE formatting guidelines.

I suggested the title should be modified as…” Exploring Knowledge, Attitudes, and Practices Regarding Preconception Care Among Reproductive-Age Individuals in Ethiopia: An Interpretive Description Study Using the Socio-Ecological and Intersectional Models”

Response :Thank you for the helpful suggestions. After careful consideration, we adopted the title suggested by Reviewer 2, as it better aligns with the study aim and research questions while improving clarity and conciseness. The revised title is: “Perspectives on Preconception Care in Ethiopia: Social, Cultural, and Structural Determinants.”

Abstract

You said that in Ethiopia, PCC remains almost absent in everyday health practice; the pooled

utilization of PCC in Ethiopia is 26%. “The magnitude of preconception care utilization

and associated factors among women in Ethiopia: systematic review and meta-analysis,

2024 | BMC Pregnancy and Childbirth”, can we say almost absent? Please justify.Similarly, you mentioned again: “This study provides the first in-depth, urban Ethiopian analysis showing how adults actively negotiate preconception care (PCC) within intersecting gender, moral, and structural systems.” However, more than three published qualitative studies on preconception care already exist in Ethiopia. As a researcher, particularly a qualitative one, honesty is essential to maintain the trustworthiness and credibility of the study

Response :Thank you for this important comment. We have revised the wording to reflect that PCC in Ethiopia is limited and inconsistently integrated into routine health services rather than “almost absent.” We have also corrected the statement to acknowledge existing qualitative studies and clarify the specific contribution of our study.

You employed maximum variation sampling, yet your study participants were drawn from the reproductive-age group (both women and men), which constitutes a relatively homogeneous population. Do you believe maximum variation sampling is appropriate for this study?.

Response :Thank you for this comment. Although participants were within the reproductive-age group, maximum variation sampling was used to capture diversity across intersecting factors such as gender, age, marital status, education, and reproductive experiences. This enabled exploration of both variability and shared patterns, consistent with the interpretive description approach. The Results were interpreted with attention to these differences and commonalities, and this has now been clarified in the Methods section.

Is interpretive description design more dominant in exploratory studies

Or Explanatory studies? Make sure that you use the appropriate design for your research.

Response :Thank you for this comment. Interpretive description is commonly used in exploratory qualitative research to generate practice-relevant insights from participants’ experiences. Our study aims to explore how reproductive-age individuals understand and experience preconception care within their social and health system context. Therefore, the use of interpretive description is appropriate for the exploratory nature of the research, and this has been clarified in the revised document

Are all the listed keywords scientifically appropriate and aligned with standard terminology? Please modify the words included in the keyword

Response :We thank the reviewer for this helpful comment. We have revised the keywords to ensure alignment with standard scientific terminology and indexing terms. Kindly see the color change in the main document.

Introduction:

What is the primary objective of this stud

---

## [Decision Letter · Decision Letter 1]

1 May 2026

PONE-D-25-65367R1Perspectives on preconception care in Ethiopia: social, cultural, and structural determinants.PLOS One

Dear Dr. Aynalem,

Thank you for submitting your manuscript to PLOS ONE. After careful consideration, we feel that it has merit but does not fully meet PLOS ONE’s publication criteria as it currently stands. Therefore, we invite you to submit a revised version of the manuscript that addresses the points raised during the review process.

**ACADEMIC EDITOR:**  General suggestion

This is a **relevant and timely qualitative study** addressing an important gap in preconception care (PCC) in Ethiopia. The manuscript is generally well-written, methodologically sound, and grounded in an appropriate qualitative framework (interpretive description). The topic aligns well with global health priorities.CommentsThe manuscript would benefit from a clearer and more explicit justification for the use of interpretive description (ID) over other qualitative approaches.While ID is appropriate for generating practice-oriented insights, the authors should clarify:Why ID is particularly suited to addressing the study objectivesHow it enables insights beyond what other qualitative designs might offerThere remains a lack of conceptual clarity regarding the study’s framing. Although the manuscript aims to provide an interpretive, socio-ecological understanding of preconception care (PCC), the research questions and parts of the analysis still reflect a KAP orientation.The authors should clearly by either:Removing explicit KAP language from the research questions, orExplicitly justifying how KAP elements are being reinterpreted through a socio-ecological and intersectional lensThis alignment is essential to ensure coherence between the study aim, methodology, and findings.The results section, while rich, remains largely descriptive and would benefit from deeper interpretive analysis.The manuscript should move beyond describing themes to explaining:How and why factors interactThe mechanisms underlying observed patternsThe relationships between themesFor example, the discussion of men’s exclusion from PCC services would be strengthened by explicitly analyzing how gender norms, health service design and institutional practices interact to produce and reinforce this exclusion.Strengthening this level of interpretation will significantly enhance the analytical contribution of the study.Integration of theoretical FrameworksAlthough the socio-ecological model and intersectionality are mentioned, their integration into the findings is not sufficiently explicit.The authors should clearly (individual, interpersonal, community, institutional) and explicitly demonstrate : how gender, marital status, age, and socioeconomic position interact to shape PCC access and experiences.This will improve conceptual coherence and demonstrate how theory informs the interpretation of results.Given that Ethiopia has already introduced a national PCC guideline (2024), the manuscript should more clearly frame its implications in terms of implementation gaps rather than suggesting the introduction of PCC.Specifically, the authors should clarify how their findings explain why PCC remains weakly integrated into routine practice and link identified barriers to failures in implementation rather than absence of policy. This reframing will enhance the policy relevance and accuracy of the study.

We look forward to receiving your revised manuscript.

Kind regards,

Dereje Haile, Ph.D

Academic Editor

PLOS One

Journal Requirements:

Additional Editor Comments:

General suggestion

This is a **relevant and timely qualitative study** addressing an important gap in preconception care (PCC) in Ethiopia. The manuscript is generally well-written, methodologically sound, and grounded in an appropriate qualitative framework (interpretive description). The topic aligns well with global health priorities.

Comments

The manuscript would benefit from a clearer and more explicit justification for the use of interpretive description (ID) over other qualitative approaches.

While ID is appropriate for generating practice-oriented insights, the authors should clarify:

Why ID is particularly suited to addressing the study objectivesHow it enables insights beyond what other qualitative designs might offer

There remains a lack of conceptual clarity regarding the study’s framing. Although the manuscript aims to provide an interpretive, socio-ecological understanding of preconception care (PCC), the research questions and parts of the analysis still reflect a KAP orientation.

The authors should clearly by either:

Removing explicit KAP language from the research questions, orExplicitly justifying how KAP elements are being reinterpreted through a socio-ecological and intersectional lens

This alignment is essential to ensure coherence between the study aim, methodology, and findings.

The results section, while rich, remains largely descriptive and would benefit from deeper interpretive analysis.

The manuscript should move beyond describing themes to explaining:

How and why factors interactThe mechanisms underlying observed patternsThe relationships between themes

For example, the discussion of men’s exclusion from PCC services would be strengthened by explicitly analyzing how gender norms, health service design and institutional practices interact to produce and reinforce this exclusion.

Strengthening this level of interpretation will significantly enhance the analytical contribution of the study.

Integration of theoretical Frameworks

Although the socio-ecological model and intersectionality are mentioned, their integration into the findings is not sufficiently explicit.

The authors should clearly (individual, interpersonal, community, institutional) and explicitly demonstrate : how gender, marital status, age, and socioeconomic position interact to shape PCC access and experiences.

This will improve conceptual coherence and demonstrate how theory informs the interpretation of results.

Given that Ethiopia has already introduced a national PCC guideline (2024), the manuscript should more clearly frame its implications in terms of implementation gaps rather than suggesting the introduction of PCC.

Specifically, the authors should clarify how their findings explain why PCC remains weakly integrated into routine practice and link identified barriers to failures in implementation rather than absence of policy. This reframing will enhance the policy relevance and accuracy of the study.

Reviewers' comments:

Reviewer's Responses to Questions

**Comments to the Author**

1. If the authors have adequately addressed your comments raised in a previous round of review and you feel that this manuscript is now acceptable for publication, you may indicate that here to bypass the “Comments to the Author” section, enter your conflict of interest statement in the “Confidential to Editor” section, and submit your "Accept" recommendation.

Reviewer #1: (No Response)

Reviewer #2: (No Response)

Reviewer #3: (No Response)

2. Is the manuscript technically sound, and do the data support the conclusions?

Reviewer #1: Yes

Reviewer #2: Yes

Reviewer #3: Yes

3. Has the statistical analysis been performed appropriately and rigorously? 

Reviewer #1: N/A

Reviewer #2: Yes

Reviewer #3: Yes

4. Have the authors made all data underlying the findings in their manuscript fully available?

Reviewer #1: No

Reviewer #2: Yes

Reviewer #3: Yes

5. Is the manuscript presented in an intelligible fashion and written in standard English?

Reviewer #1: Yes

Reviewer #2: No

Reviewer #3: Yes

6. Review Comments to the Author

Reviewer #1: Overall, the author's response was good; however, the issues raised—especially in the results section—were not addressed adequately. These issues still need to be responded to by the author.

1. The overall presentation of the results seems shallow, incoherent, and vague. The data likely requires reanalysis, and the presentation of the themes and subthemes needs to be updated accordingly.

2. You submitted the PARTICIPANT CONSENT FORM in Supplementary File 1, but what I require is the actual form that the participants signed. This is crucial information for establishing the trustworthiness of the overall results.

3. I still do not agree with the listed keywords. You have provided the keywords according to scientific conventions, including their order.

Reviewer #2: The manuscript addresses an important topic and presents rich qualitative data, but several methodological, analytical, and reporting issues must be addressed before this paper can be considered for publication. Major revisions are required to clarify sampling, analytic rigor, conceptual framing, and the linkage between findings and clinical or policy implications.

Reviewer #3: The manuscript presents a rigorous and well-conducted qualitative study. The methodology is appropriate, with clear description of data collection and analysis procedures, and the findings are well supported by the data. The interpretations are credible and aligned with the results. The manuscript is clearly written in standard English and meets the journal’s requirements. Overall, it is suitable for publication after minor revisions.

7. PLOS authors have the option to publish the peer review history of their article (what does this mean?). If published, this will include your full peer review and any attached files.

Reviewer #1: No

Reviewer #2: No

Reviewer #3: No

---

## [Author Response · Author response to Decision Letter 2]

4 May 2026

Authors Response

Editor and Reviewer Comments Comments/ Revisions Made

Editor comment

General suggestion

This is a relevant and timely qualitative study addressing an important gap in preconception care (PCC) in Ethiopia. The manuscript is generally well-written, methodologically sound, and grounded in an appropriate qualitative framework (interpretive description). The topic aligns well with global health priorities.

Thank you, Dear Editor, for your positive and encouraging feedback.

Comments

The manuscript would benefit from a clearer and more explicit justification for the use of interpretive description (ID) over other qualitative approaches.

• While ID is appropriate for generating practice-oriented insights, the authors should clarify:

• Why ID is particularly suited to addressing the study objectives

• How it enables insights beyond what other qualitative designs might offer

Dear Editor,

Thank you for your feedback. We have revised the manuscript accordingly. Please also note that this paper provides only a brief introduction and methodological overview, as it is part of a larger dissertation project. The full study includes healthcare workers, stakeholders, and reproductive-age individuals, and a dedicated dissertation chapter presents detailed methods, including the rationale for selecting interpretive description over other qualitative approaches. We encourage interested readers to refer to that for a more comprehensive account in the near future .

• There remains a lack of conceptual clarity regarding the study’s framing. Although the manuscript aims to provide an interpretive, socio-ecological understanding of preconception care (PCC), the research questions and parts of the analysis still reflect a KAP orientation.

Thank you for this comment. We respectfully note that this was addressed in our R1 revision. The research question has been revised to focus on in-depth perspectives and experiences rather than discrete knowledge or attitudes, consistent with ID's interpretive focus. While knowledge, attitudes, and practices emerged as part of participants' experiences, the analysis interpreted these within socio-ecological and intersectional contexts. The aim has been further clarified in the revised manuscript to ensure alignment with the title, analytic approach, and conclusions; please see the highlighted changes

• The authors should clearly by either:

• Removing explicit KAP language from the research questions, or

• Explicitly justifying how KAP elements are being reinterpreted through a socio-ecological and intersectional lens

• This alignment is essential to ensure coherence between the study aim, methodology, and findings.

Thank you for this comment. We respectfully note that this was addressed in our R1 revision. We have nonetheless conducted a thorough review of the manuscript to ensure full alignment; please see the highlighted changes."

Result

The results section, while rich, remains largely descriptive and would benefit from deeper interpretive analysis.

The manuscript should move beyond describing themes to explaining:

How and why factors interact

The mechanisms underlying observed patterns

The relationships between themes

For example, the discussion of men’s exclusion from PCC services would be strengthened by explicitly analyzing how gender norms, health service design and institutional practices interact to produce and reinforce this exclusion.

Strengthening this level of interpretation will significantly enhance the analytical contribution of the study.

Thank you for this suggestion. We have addressed it; please see the changes highlighted in the revised manuscript.

Integration of Theoretical Frameworks

Although the socio-ecological model and intersectionality are mentioned, their integration into the findings is not sufficiently explicit.

The authors should clearly (individual, interpersonal, community, institutional) and explicitly demonstrate : how gender, marital status, age, and socioeconomic position interact to shape PCC access and experiences.

This will improve conceptual coherence and demonstrate how theory informs the interpretation of results.

Thank you for this comment. As clarified in the manuscript, the socio-ecological model and intersectionality informed interpretation rather than structuring the analysis a priori. We have strengthened the presentation to more explicitly demonstrate how factors across levels and intersecting identities shape PCC experiences in the findings.

Given that Ethiopia has already introduced a national PCC guideline (2024), the manuscript should more clearly frame its implications in terms of implementation gaps rather than suggesting the introduction of PCC.

Specifically, the authors should clarify how their findings explain why PCC remains weakly integrated into routine practice and link identified barriers to failures in implementation rather than absence of policy. This reframing will enhance the policy relevance and accuracy of the study

Thank you. We have revised it.

Reviewer's Responses to Questions

1. If the authors have adequately addressed your comments raised in a previous round of review and you feel that this manuscript is now acceptable for publication, you may indicate that here to bypass the “Comments to the Author” section, enter your conflict of interest statement in the “Confidential to Editor” section, and submit your "Accept" recommendation.

Reviewer #1: (No Response)

Reviewer #2: (No Response)

Reviewer #3: (No Response)

-

2. Is the manuscript technically sound, and do the data support the conclusions?

Reviewer #1: Yes

Reviewer #2: Yes

Reviewer #3: Yes We thank the reviewers for confirming the technical rigor and that the data support the conclusions.

3. Has the statistical analysis been performed appropriately and rigorously?

Reviewer #1: N/A

Reviewer #2: Yes

Reviewer #3:Yes Thank you for your feedback.

4. Have the authors made all data underlying the findings in their manuscript fully available?

Reviewer #1: No

Reviewer #2: Yes

Reviewer #3:Yes Thank you for the feedback. We have now fully complied with the PLOS Data policy. All underlying data have been provided as Supporting Information, and the Data Availability Statement has been updated accordingly.

5. Is the manuscript presented in an intelligible fashion and written in standard English?

Reviewer #1: Yes

Reviewer #2: No

Reviewer #3:Yes Thank you for your feedback. The manuscript has been thoroughly revised to improve grammar, clarity, and overall readability.

Review Comments to the Author

Reviewer 1 Comments to the Author

Overall, the author's response was good; however, the issues raised—especially in the results section—were not addressed adequately. These issues still need to be responded to by the author.

Thank you for your feedback. We sincerely appreciate it and have made further revisions accordingly.

The overall presentation of the results seems shallow, incoherent, and vague. The data likely requires reanalysis, and the presentation of the themes and subthemes needs to be updated accordingly.

Thank you for your comment. We believe the study robustly addresses the research questions and fully adheres to ID expectations. Nonetheless, to further satisfy the reviewer, we have refined the analysis and updated the themes and subthemes to improve clarity, coherence, and depth; please see the highlighted changes in the revised manuscript.

You submitted the PARTICIPANT CONSENT FORM in Supplementary File 1, but what I require is the actual form that the participants signed. This is crucial information for establishing the trustworthiness of the overall results

Thank you for your feedback and concern. Due to ethical and confidentiality considerations, we are unable to upload the signed participant consent forms. However, ethical approval details are provided in the manuscript, and the relevant ethics board can be contacted if verification is required.

I still do not agree with the listed keywords. You have provided the keywords according to scientific conventions, including their order

Thank you for your comment. We have revised it.

Reviewer 2 Comments to the Author

The manuscript addresses an important topic and presents rich qualitative data, but several methodological, analytical, and reporting issues must be addressed before this paper can be considered for publication. revisions are required to clarify sampling, analytic rigor, conceptual framing, and the linkage between findings and clinical or policy implications Dear Reviewer,

Thank you for your feedback. We have revised the manuscript to address sampling, analytic rigor, conceptual framing, and clinical and policy implications.

Framing, citations, and epidemiological currency

Citation over-reliance: The manuscript appears to rely heavily on a single citation ([25]) to frame the core problem statement. Please broaden and update the literature review to include more recent and regionally relevant studies (local Ethiopian and sub-Saharan African PCC literature) to situate your contribution within the current discourse. Thank you for this comment. We have revised it accordingly.

Outdated epidemiological data: The maternal mortality and other public-health statistics cited seem outdated. Please update these figures to the latest sources (e.g., recent EDHS, WHO/UNICEF 2024 updates) and justify your choice of statistics to ensure clinical and policy relevance. Dear Reviewer,

Thank you for this comment. We have updated the epidemiological data using recent sources.

Sampling, representativeness, and generalizability

Sample size and representativeness: The study was conducted in a single capitalcity setting with only 18 participants. Please clarify that findings are not nationally representative and discuss transferability limits. Avoid implying national generalizability based on this urban, hospital-based sample.

Sample composition and gender split: Provide methodological justification for the 10 women / 8 men composition. Was this ratio purposeful, convenience-based, or determined by saturation? Explain how you assessed saturation for each gender subgroup, and justify claims (e.g., “men’s latent desire”) given the small male sample. Thank you for these points. We have addressed both concerns in the limitations section and clarified that the study assessed information power rather than data saturation, based on the study aim, sample specificity, narrative depth, and the interpretive descriptive approach. We respectfully note that these issues were addressed in detail in our R1 revision, and we kindly request that the reviewer refer to the revised manuscript before raising further concerns on these points; please see the highlighted changes for full details

Methods and design choices

Choice of Interpretive Description (ID): Explain why ID was selected over other qualitative frameworks (e.g., Grounded Theory, Phenomenology). Specifically justify how the applied nature of ID produced clinically relevant insights that a traditional thematic or descriptive approach could not. Thank you for this comment. We have clarified the rationale for selecting interpretive description. Please see the highlighted changes for full details

Research questions and methodological alignment: o KAP framing vs. ID: Research Question 1 is framed in KnowledgeAttitudes-Practices (KAP) terms, which aligns poorly with ID’s interpretive, practice-oriented focus. Explain why a KAP-style question was used and how it allows for the depth of interpretation expected in ID. Thank you for this comment. We respectfully note that this issue was addressed in our R1 revision, where the research question was revised to focus on in-depth perspectives and experiences rather than discrete knowledge or attitudes, strengthening alignment with the interpretive, practice-oriented focus of ID. We kindly request that the reviewer refer to the updated manuscript; please see the highlighted changes for full details

Passive phrasing: RQ1’s wording (“What can be learned...”) is passive and undercuts the applied focus of ID. Consider rephrasing to emphasize processes of negotiation or construction of PCC in context. Thank you for this comment. The research questions have been revised to ensure active wording and better alignment with the applied focus of interpretive description.

Clinical utility: ID prioritizes clinical relevance. Neither RQ explicitly asks about implications for service delivery, nursing, or midwifery practice. Consider adding or revising an RQ to directly probe actionable pathways for health-system adaptation. Thank you for this suggestion. We note that this study forms part of a larger qualitative interpretive project exploring PCC across multiple groups, including healthcare workers and stakeholders, with findings collectively oriented toward actionable insights for service delivery and nursing and midwifery practice. We have revised the research questions to more explicitly reflect this; please see the highlighted changes.

Analytic rigor and reflexivity

Transcription and translation: Provide details on transcription and translation processes (who transcribed/translated, whether translations were back-translated or verified, and steps taken to preserve meaning). Dear Reviewer,

Thank you. We have clarified the transcription and translation procedures, including verification and meaning preservation.

Coding and analytic procedures: Describe the coding team, development and use of a codebook, procedures for inter-coder checking, and how themes were iteratively developed. If only one analyst, detail measures used to enhance credibility (peer debriefing, memoing, triangulation). Thank you. We have clarified the coding process, team roles, and steps taken to enhance analytic rigor, including iterative development and verification procedures.

Saturation and thematic adequacy: Explain how you assessed sufficiency of data and reached saturation for the reported seven themes given 18 interviews.

Thank you for this comment. As addressed in R1, sample adequacy was guided by information power, and we have clarified how this supports the development of the reported themes.

Reflexivity: Add a reflexivity statement describing researcher positionality (interviewer gender, language skills, professional roles), relationships with participants, and how potential biases were managed.

Thank you. We have added a reflexivity statement outlining researcher positionality, relationships with participants, and steps taken to manage potential biases.

Thematic presentation and conceptual clarity

Evidence and quotes: Strengthen each theme with clearer definitions and representative verbatim quotes (with consistent pseudonyms and key participant characteristics: gender, age, marital status). For strong claims (e.g., “institutional exclusion of unmarried women”), clearly delineate whether evidence reflects formal policy, provider attitudes, or actual practices. Thank you. This has already been addressed in the revised manuscript. See supple

---

## [Decision Letter · Decision Letter 2]

24 May 2026

Perspectives on preconception care in Ethiopia: social, cultural, and structural determinants.

PONE-D-25-65367R2

Dear Dr. Yared Asmare Aynalem,

We’re pleased to inform you that your manuscript has been judged scientifically suitable for publication and will be formally accepted for publication once it meets all outstanding technical requirements.

Kind regards,

Dereje Haile, Ph.D

Academic Editor

PLOS One

Additional Editor Comments (optional):

Reviewers' comments:

Reviewer's Responses to Questions

**Comments to the Author**

1. If the authors have adequately addressed your comments raised in a previous round of review and you feel that this manuscript is now acceptable for publication, you may indicate that here to bypass the “Comments to the Author” section, enter your conflict of interest statement in the “Confidential to Editor” section, and submit your "Accept" recommendation.

Reviewer #1: (No Response)

Reviewer #3: All comments have been addressed

2. Is the manuscript technically sound, and do the data support the conclusions?

Reviewer #1: Yes

Reviewer #3: Yes

3. Has the statistical analysis been performed appropriately and rigorously? 

Reviewer #1: N/A

Reviewer #3: N/A

4. Have the authors made all data underlying the findings in their manuscript fully available?

Reviewer #1: Yes

Reviewer #3: Yes

5. Is the manuscript presented in an intelligible fashion and written in standard English?

Reviewer #1: Yes

Reviewer #3: Yes

6. Review Comments to the Author

Reviewer #1: The primary purpose of peer review comments on a manuscript is to enhance its quality prior to publication. However, the authors have not revised the Results section as suggested

Reviewer #3: Congratulations to all authors, you have addressed all comments. This research has potential of filling knowledge factors in the area. Considering this I declare Accept for publication. Congratulations to all authors, you have addressed all comments. This research has potential of filling knowledge factors in the area. Considering this I declare Accept for publication.

7. PLOS authors have the option to publish the peer review history of their article (what does this mean?). If published, this will include your full peer review and any attached files.

Reviewer #1: **Yes:** Gebremedhin Gebreegziabher Gebretsadik

Reviewer #3: No

---

## [Editor Report · Acceptance letter]

PONE-D-25-65367R2

PLOS One

Dear Dr. Aynalem,

I'm pleased to inform you that your manuscript has been deemed suitable for publication in PLOS One. Congratulations! Your manuscript is now being handed over to our production team.

Kind regards,

on behalf of

Dr. Dereje Haile

Academic Editor

PLOS One